# An Intelligent Fault Diagnosis Method Using GRU Neural Network towards Sequential Data in Dynamic Processes

**Jing Yuan [1] and Ying Tian [2],***

[1]  School of Mechanical Engineering, University of Shanghai for Science and Technology, Shanghai 200093, China; 173780670@st.usst.edu.cn

[2]  School of Optical-Electrical and Computer Engineering, University of Shanghai for Science and Technology, Shanghai 200093, China

*  Correspondence: tianying@usst.edu.cn; Tel.: +86-131-2097-5286

**Abstract:** Intelligent fault diagnosis is a promising tool to deal with industrial big data due to its ability in rapidly and efficiently processing collected signals and providing accurate diagnosis results. In traditional static intelligent diagnosis methods, however, the correlation between sequential data is neglected, and the features of raw data cannot be effectively extracted. Therefore, this paper proposes a three-stage fault diagnosis method based on a gate recurrent unit (GRU) network. The raw data is divided into several sequence units by first using a moving horizon as the input of GRU. In this way, we can intercept the sequence to get information as needed. Then, the GRU deep network is established through batch normalization (BN) algorithm to extract the dynamic feature from the sequence units effectively. Finally, the softmax regression is employed to classify faults based on dynamic features. Thus, the diagnosis result is obtained with a probabilistic explanation. Two chemical processes validate the proposed method: Tennessee Eastman (TE) benchmark process as well as para-xylene (PX) oxidation process. In the case of TE, the diagnosis results demonstrate the proposed method is superior to conventional methods. Furthermore, in the case of PX oxidation, the result shows that the proposed method also has an exceptional effect with a little historical data.

**Keywords:** dynamic process; fault diagnosis; gate recurrent unit (GRU); moving horizon

## 1. Introduction

With the advancement of modern industrial technology and process control mechanisms, an industrial process has become more and more complex [1,2]. To improve the industry process safety and product quality, process monitoring and fault diagnosis have received lots of attention over the past few decades [3]. Data-driven multivariate statistical process monitoring (MSPM) has been widely applied to the monitoring of industrial process operations and production results. Compared to knowledge-based methods and model-based methods, MSPM methods are more accessible to establish with less or even no demand of the accurate kinematic equations [4,5]. As a result, MSPM models, such as principal component analysis (PCA) and independent component analysis (ICA), are widely used in industrial process monitoring and fault diagnosis [6].

Traditionally, the framework of fault diagnosis includes two main steps: (1) feature extraction; and (2) fault classification. In the feature extraction step, many methods have been proposed to map the raw data from the high-dimensional space into a low-dimensional feature space, and then perform fault diagnosis in that feature space. The PCA, ICA, partial least squares (PLS), and linear discriminant analysis (LDA) are the most widely used feature extraction methods in the fields of fault diagnosis. In the second step, various classifiers, such as neural networks of multi-layer perceptron (MLP) [7],

support vector machine (SVM) [8], Bayesian discriminant functions [9], and adaptive neuro-fuzzy inference system (ANFIS) [10], have been applied for fault classification. "Feature extraction + classification" fault diagnosis strategies like PCA + SVM and ICA + MLP have obtained satisfactory results. However, static modelling methods like PCA and LDA assume that data samples are collected independently from sensors without sequence correlation. It is well known that most industrial processes evolve from past operation situations to potential future events [11]. Therefore, dynamic behavior should be one of the essential characteristics of industrial process data [12]. In order to extract the dynamic features of the sequence data, dynamic principal component analysis (DPCA) [13] and dynamic linear discriminant analysis (DLDA) [14], among others, has been developed by augmenting each measurement with a fixed length of several previous measurements and aligned to a stacking matrix [15]. Some fault diagnosis methods for the dynamic process like DPCA-SVM and DLDA-SVM have been developed. However, conventional methods still have some obvious drawbacks as follows:

(1) Vector-based augmentation may aggravate the "curse of dimensionality" problem and make the feature extraction methods unstable [16,17].

(2) Feature extraction and classification both affect the diagnosis performance but are designed individually. This is a divide and conquer strategy that cannot be optimized simultaneously.

(3) The extracted features are usually hand-crafted, requiring much prior knowledge about process monitoring techniques and diagnostic expertise, which is time-consuming and labor-intensive.

With the rapid advancement of machine learning, deep learning has developed as an efficient way to overcome the above drawbacks. Deep learning can learn the abstract representation features of the raw data automatically, which could avoid the requirement of prior knowledge. Deep learning is a branch of machine learning algorithms that attempt to model complexity and internal correlation in a dataset by using multiple processing layers, or with complex structures, to mine the information hidden in the dataset for classification or other goals [18]. In recent years, deep learning has developed rapidly in academic and industrial fields. Tang et al. applied deep belief networks (DBNs) to fault feature extraction and diagnosis of the chemical industry and introduced the quadratic programming method to estimate the sparse coefficients simultaneously class by class [18]. Wen et al. convert fault signals into two-dimensional (2-D) images and adopt convolutional neural networks (CNNs) to extract the features of the converted 2-D images [19]. However, the above methods are all static network applications. Hochreiter et al. proposed recurrent neural networks (RNNs) [20]. An RNN is more suitable for fault diagnosis of dynamic processes because an RNN takes full account of the associations among samples. This association is represented by the connection of neurons in the RNN's hidden layer. You et al. adopted an RNN to diagnose battery states in electric vehicle systems and determine the replacement time for a battery or to assess the driving mileage [21].

Gated recurrent unit (GRU) [22], a variant of RNN, not only retains all the advantages of RNN but also adds "gate" operations to its hidden layer neurons, which allows GRU to maintain useful information and discard useless information in dynamic sequence data automatically. A GRU demonstrates state-of-the-art performance on sequential problems including natural language processing, image classification, and time series prediction. For the purpose of diagnosing the faults of dynamic process accurately, quickly, and effectively, this paper proposes a three stage fault diagnosis method-based GRU deep network. The main contributions of this paper are as follows:

(1) Following the fault diagnosis framework, we propose a three-stage method. In the first stage, a moving horizon is adopted to process dynamic process data such that raw data is entered into the GRU without losing any dynamic information. In the second stage, we apply the GRU deep network belonging to deep learning to the extract the dynamic feature of sequential data. Moreover, in the third stage, softmax regression is adopted to obtain the output with a probabilistic explanation.

(2) Two diagnostic case studies were used to validate the proposed method. In the Tennessee Eastman (TE) case, the parameter selection of the method was studied in depth. Furthermore, the proposed method is compared to the conventional methods. The comparison results show the superiority

of the method. In the case of para-xylene (PX) oxidation process, the diagnosis results show that the method can be easily and effectively applied to other diagnostic problems.

(3)  Considering the covariate shift in deep learning and the over-fitting caused by the "curse of dimensionality," BN is applied to our method to reduce the training time of GRU and improve the accuracy of fault diagnosis.

This paper is organized as follows. In Section 2, a simple RNN and its variant GRU are introduced in detail. Meanwhile, batch normalization and softmax regression are briefly described. Section 3 details the proposed three-stage learning method. In Sections 4 and 5, the efficiency and accuracy of the proposed method are illustrated in the TE process as well as the PX oxidation process. Finally, the conclusion is provided in Section 6.

## 2. Recurrent Neural Network and Softmax Regression

### 2.1. Concept of an RNN

An RNN is called recurrent because they perform the same task for each element in the sequence. The RNN uses the hidden state to record the state of each moment while processing the sequence data, and the current state depends on the current input as well as the state of the previous moment. Therefore, the current hidden state makes full use of past information. In this way, an RNN can process sequence data in dynamic processes. The architecture of an RNN is shown in Figure 1. When given an input sequence $X = [x_1, x_2 \ldots x_t \ldots x_T]$ of length $T$, an RNN defines the hidden state $h_t$ at the time $t$ of a sequence as:

$$h_t = \tanh(W_h h_{t-1} + W_x x_t + b) \tag{1}$$

where $W_h \in \mathbb{R}^{d_h \times d_h}$ is the weight matrix between hidden layers, $W_x \in \mathbb{R}^{d_h \times d_x}$ is the weight matrix of the input layer to the hidden layer, and $b \in \mathbb{R}^{d_h}$ is the bias. $W_h$, $W_x$, $b$, and the initial state $h_0 \in \mathbb{R}^{d_h}$ are parameters of the RNN. The tanh is the activation function.

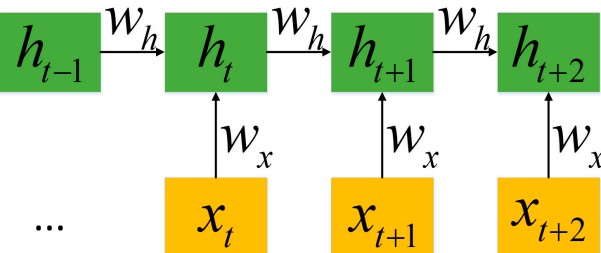

**Figure 1.** The architecture of an RNN.

Although the RNN is very powerful when dealing with sequence problems, it is difficult to train with the gradient descent method because of the well-known gradient vanishing/explosion problem [20]. On the other hand, variants of RNN have been developed to solve the above problems, such as Long Short-Term Memory (LSTM), GRU, etc. Among them, GRU avoids overfitting, as well as saves training time. Therefore, GRU is adopted in our method.

### 2.2. Concept of a GRU

GRU has the same chain structure as a simple RNN, but a GRU is more complicated in the way it updates the hidden state. Instead of directly updating the current hidden state with the previous hidden state, GRU uses a reset gate and updates the gate, which can judge whether the information in the previous hidden state is useful, then holds useful information and removes useless information. Figure 2 shows the architecture of GRU. The way GRU updates $h_t$ is as follows:

(1)  The reset gate $r_t$ and update gate $z_t$:

$$z_t = \sigma(W_{zh} h_{t-1} + W_{zx} x_t + b_z) \tag{2}$$

$$r_t = \sigma(W_{rh}h_{t-1} + W_{rx}x_t + b_r) \tag{3}$$

The activation function $\sigma$ is the sigmoid function, and the value range of each element in the reset gate $r_t$ and the update gate $z_t$ are [0, 1].

(2)　Candidate hidden state:

$$\widetilde{h}_t = \tanh(W_{\widetilde{h}h}(r_t * h_{t-1}) + W_{\widetilde{h}x}x_t + b_h) \tag{4}$$

The candidate hidden state $\widetilde{h}_t$ uses the reset gate $r_t$ to control the inflow of the previous hidden state $h_{t-1}$ containing past information. If the reset gate is approximately zero, the previous hidden state will be removed. Therefore, the reset gate provides a mechanism to remove previous hidden states that are unrelated to the future; that is, the reset gate determines how much information was forgotten in the past.

(3)　Hidden state:

$$h_t = z_t * h_{t-1} + (1 - z_t) * \widetilde{h}_t \tag{5}$$

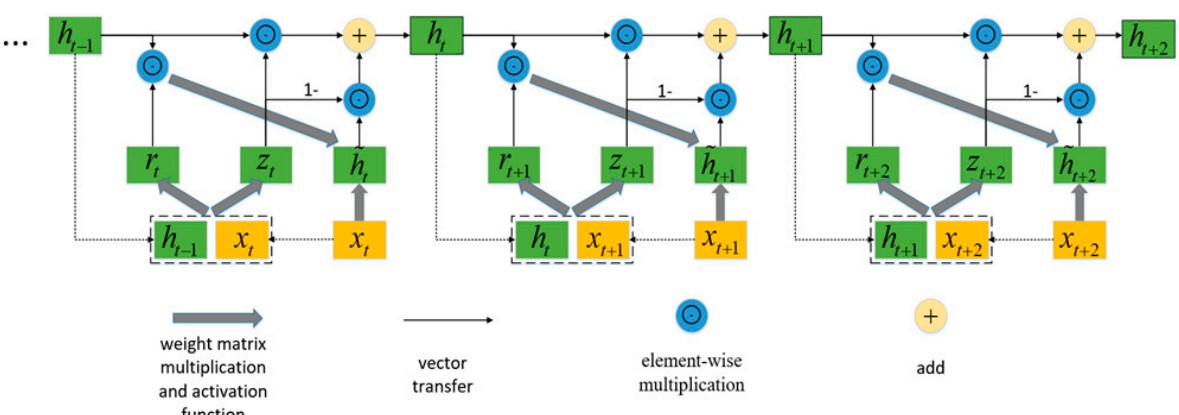

**Figure 2.** The architecture of GRU.

The hidden state $h_t$ uses the update gate $z_t$ to update the previous hidden state $h_{t-1}$ and the candidate hidden state $\widetilde{h}_t$. If the update gate is approximately 1, the previous hidden state will be held and passed to the current moment. When given an input sequence $X = [x_1, x_2 \ldots x_t \ldots x_T]$ of length $T$, GRU passes the last hidden state $h_T$ through a nonlinear transformation as the output.

$$o = \sigma(W_o h_T + b_o) \tag{6}$$

In the above formula $W_{zh}, W_{rh}, W_{\widetilde{h}h} \in \mathbb{R}^{d_h \times d_h}$ are the weight matrices of the hidden layer to the hidden layer, $W_{zx}, W_{rx}, W_{\widetilde{h}x} \in \mathbb{R}^{d_h \times d_x}$ are the weight matrices of the input layer to the hidden layer, $W_o \in \mathbb{R}^{d_o \times d_h}$ is the weight matrix of the output layer and $b_z, b_r, b_h, b_o \in \mathbb{R}^{d_h}$ are the bias $W_{zh}, W_{rh}, W_{\widetilde{h}h}, W_{zx}, W_{rx}, W_{\widetilde{h}x}, b_z, b_r, b_h, b_o$, and the initial states $h_0 \in \mathbb{R}^{d_h}$ are the parameters of the GRU.

The GRU can cope with the gradient vanishing/explosion problem in the RNN, so it is more suitable for the fault diagnosis of dynamic processes.

*2.3. Batch Normalization-Based GRU*

It is known that for deep neural networks, an internal covariate shift is a common phenomenon where the features presented to a networks change in distribution during the process of training [23]. When using a GRU that resembles very deep feed-forward networks to process sequence data for dynamic processes, this internal covariate shift may play an especially important role. In order to reduce internal covariate shift, batch normalization was proposed recently. Batch normalization involves standardizing the activations going into each layer, enforcing their means $\mu$ and variances $\sigma^2$

to be invariant to changes in the parameters of the underlying layers, so as to accelerate the training. Indeed, GRU network strained with batch normalization converge significantly faster and generalize better. The batch normalizing transform is as follows:

$$BN(c_i; \gamma, \beta) = \gamma * \frac{c_i - \mu_B}{\sqrt{\sigma_B^2 + \varepsilon}} + \beta \tag{7}$$

where $c_i \in \mathbb{R}^d$ is the vector that will be normalized, $\gamma \in \mathbb{R}^d$ and $\beta \in \mathbb{R}^d$ are model parameters that determine the mean and standard deviation of the normalized activation, and $\varepsilon \in \mathbb{R}^d$ is a regularization hyperparameter. The $*$ denotes the Hadamard product (element-wise multiplication). According to Reference [24] we set $\beta$ and $\varepsilon$ equal 0. At training time, we use the mini-batch training strategy, which divides all training samples into many mini-batches, and each mini-batch carries out a parameter update. Therefore, the input of BN is the current mini-batch containing $k$ samples, which can be expressed as $B = \{c_{1...k}\}$. $\mu_B \leftarrow \frac{1}{k} \sum_{i=1}^{k} c_i$ is the sample mean and $\sigma_B^2 \leftarrow \frac{1}{k} \sum_{i=1}^{k} (c_i - \mu_B)^2$ is the sample variance.

We introduce the batch-normalizing transform into the GRU network. Batch normalization is adopted in the hidden-to-hidden transformations as follows:

$$z_t = \sigma(BN(W_{zh}h_{t-1}) + W_{zx}x_t + b_z) \tag{8}$$

$$r_t = \sigma(BN(W_{rh}h_{t-1}) + W_{rx}x_t + b_r) \tag{9}$$

$$\widetilde{h}_t = \tanh(BN(W_{\widetilde{h}h}(r_t * h_{t-1})) + W_{\widetilde{h}x}x_t + b_h) \tag{10}$$

$$h_t = z_t * h_{t-1} + (1 - z_t) * \widetilde{h}_t \tag{11}$$

$$o = \sigma(W_o h_T + b_o) \tag{12}$$

*2.4. Softmax Regression*

In neural networks, softmax regression is often implemented at the final layer for multiclass classification. It is computed fast and can provide a result with a probabilistic explanation. Suppose that we have a training set $\{X^{(i)}\}_{i=1}^{m}$ with its label $\{Y^{(i)}\}_{i=1}^{m}$ where $X^{(i)}$ is the input sample and $Y^{(i)} \in \{1, 2, \ldots j \ldots, K\}$ is the label. It should be noted here that one should not confuse the input of softmax with the input of GRU. In fact, in our task, the input sample $X^{(i)}$ here is the output $o^{(i)}$ of the GRU network. For each input sample $X^{(i)}$, the model works to estimate the probability $P(Y^{(i)} = j | X^{(i)})$ for each label of $j = 1, 2, \ldots, K$. Thus, the result of softmax regression will output a vector that gives $K$ estimated probabilities of the input sample $X^{(i)}$ belonging to each label. Concretely, the result of softmax regression $\phi_\theta(X^{(i)})$ takes the form:

$$\phi_\theta(X^{(i)}) = \begin{bmatrix} p(Y^{(i)} = 1 | X^{(i)}; \theta) \\ p(Y^{(i)} = 2 | X^{(i)}; \theta) \\ \vdots \\ p(Y^{(i)} = K | X^{(i)}; \theta) \end{bmatrix} = \frac{1}{\sum\limits_{j=1}^{K} \exp(\theta_j^T \cdot X^{(i)})} \begin{bmatrix} \exp(\theta_1^T X^{(i)}) \\ \exp(\theta_2^T X^{(i)}) \\ \vdots \\ \exp(\theta_K^T X^{(i)}) \end{bmatrix} \tag{13}$$

where $\theta = [\theta_1, \theta_2, \ldots, \theta_K]^T$ are the parameters of the softmax regression model. It should be noticed that the term $\sum\limits_{j=1}^{K} \exp(\theta_j^T \cdot X^{(i)})$ normalizes the distribution such that the sum of the elements of result equals 1.

*2.5. Loss Function and Optimizer*

Based on the result, the whole model is trained by minimizing the cost function $J(\Theta)$:

$$J(\Theta) = -\frac{1}{m}\left[\sum_{i=1}^{m}\sum_{j=1}^{K} 1\{Y^{(i)} = j\}\log(p(Y^{(i)} = j \mid X^{(i)}; \Theta))\right] \tag{14}$$

where $\Theta = \{W_{zh}, W_{zx}, W_{rh}, W_{rx}, W_{\tilde{h}h}, W_{\tilde{h}x}, W_o, b_z, b_r, b_h, b_o, h_0, \gamma, \theta\}$ is the set of parameters containing all the parameters above. As mentioned earlier, this article uses the mini-batch training strategy, so $m$ here can be understood as a mini-batch. Furthermore, in the experiments in Sections 4 and 5, the setting of the mini-batch will be given. $K$ is the number of classes, $1\{Y^{(i)} = j\}$ is the indicator function indicating that if the class of the *i*th sample is *j*, then $1\{Y^{(i)} = j\} = 1$, otherwise $1\{Y^{(i)} = j\} = 0$.

In this paper, we use Adam to optimize the loss function. Adam is a first-order optimization algorithm that can replace the traditional stochastic gradient descent process. It can iteratively update neural network parameters based on training data. The stochastic gradient descent maintains a single learning rate to update all parameters, and the learning rate does not change during the training process. Adam calculates independent adaptive learning rates for different parameters by calculating the first-moment estimation and second-moment estimation of the gradient. The pseudocode of the Adam algorithm for updating $\Theta$ is shown in Figure 3. For more details regarding Adam, please refer to Reference [25].

---

- $\alpha = 0.001, \beta_1 = 0.9, \beta_2 = 0.999, \eta = 10^{-8}$ (Defaults)

$m_0 \leftarrow 0$ (Initialize 1st moment vector)

$v_0 \leftarrow 0$ (Initialize 2nd moment vector)

$i \leftarrow 0$ (Initialize step)

**while** $\Theta_i$ not converged **do**

    $i \leftarrow i + 1$

    $g_i \leftarrow \nabla_{\Theta} f_i(\Theta_{i-1})$ (Get gradients at step $i$)

    $m_i \leftarrow \beta_1 \cdot m_{i-1} + (1 - \beta_1) \cdot g_i$ (Update biased first moment estimate)

    $v_i \leftarrow \beta_2 \cdot v_{i-1} + (1 - \beta_2) \cdot g_i^2$ (Update biased second raw moment estimate)

    $\hat{m}_i \leftarrow m_i / (1 - \beta_1^i)$ (Compute bias-corrected first moment estimate)

    $\hat{v}_i \leftarrow v_i / (1 - \beta_2^i)$ (Compute bias-corrected second raw moment estimate)

    $\Theta_i \leftarrow \Theta_{i-1} - \alpha \cdot \hat{m}_i / (\sqrt{\hat{v}_t} + \eta)$ (Update parameters)

**end while**

**return** $\Theta_i$ (resulting parameters)

---

**Figure 3.** Pseudocode of the Adam algorithm.

## 3. Three-Stage Fault Diagnosis Method of Dynamic Process

This section details the proposed three-stage fault diagnosis method for fault diagnosis of the dynamic process. The illustration and flowchart of the method are shown in Figure 3. In the first stage, the moving horizon was used to process raw data as the input sequences of GRU. In the second stage, the GRU model was established through batch normalization, and the model was trained with sequences processed by moving horizon. In this way, the GRU model extracts the dynamic features in the raw data. In the third stage, softmax regression was applied to classify faults using the extracted dynamic features.

*3.1. First Stage—Moving Horizon*

In order to make full use of the correlation among sequential data of the dynamic process, we adopted the moving horizon to process the raw data. The width of the moving horizon can be

adjusted according to different needs. The width of the moving horizon is the length of the input sequence which is defined as time steps (T) in GRU. For example, suppose there are $n$ sets of raw data $X = [x_1, x_2 \ldots x_n]$ where $x \in \mathbb{R}^{d_x \times 1}$, when the time steps are set to 3 ($T$ = 3), then the moving horizon divides raw data into several sequences like $[x_1, x_2, x_3][x_2, x_3, x_4][x_3, x_4, x_5]$... such that there are $m = n - T$ sequences, and each sequence is an input sample to the GRU neural network.

### 3.2. Second Stage—Extract Dynamic Features by GRU

Once the input sequences of GRU is obtained, we define the input in this way $X^{(1)} = [x_1, x_2, x_3]$, $X^{(2)} = [x_2, x_3, x_4], \ldots, X^{(m)} = [x_{n-2}, x_{n-1}, x_n]$, where $x \in \mathbb{R}^{d_x \times 1}$ and $X \in \mathbb{R}^{d_x \times 3}$. What needs to be explained here is that we just use $T$ = 3 as an example. In fact, the time step can be adjusted according to different needs. Each input $X^{(i)}$ corresponds to an output $o^{(i)}$ refers to Equations (2), (3), (4), (5), and (6). During this time, the output vector $o^{(i)}$ is the dynamic features extracted by GRU.

### 3.3. Third Stage—Obtain Fault Diagnosis Result Using Softmax Regression

This section details the proposed three-stage fault diagnosis method for fault diagnosis of the dynamic process. The illustration and flowchart of the method are shown in Figure 4. In the first stage, the moving horizon is used to process raw data as the input sequences of GRU. In the second stage, the GRU model is established through batch normalization, and the model is trained with sequences processed using a moving horizon. In this way, the GRU model extracts the dynamic features in the raw data. In the third stage, softmax regression is applied to classify faults using the extracted dynamic features. Once the dynamic features set $\{o^{(i)}\}_{i=1}^{m}$ is obtained, we combined it with the label set $\{Y^{(i)}\}_{i=1}^{m}$ to train the softmax regression. The softmax regression model computes the probability that the feature $o^{(i)}$ has the fault labels $Y^{(i)}$ as in Equation (12). The sum of the probabilities over all class labels being 1 ensures that the right side in Equation (12) defines a properly normalized distribution. After being trained, the maximum posterior probability in $\phi_\theta(o^{(i)})$ indicates which fault label the feature $o^{(i)}$ belongs to.

After the three stages, we used test samples to verify the proposed method. For example, there were new samples of a dynamic process $X^{new} = [x_1^{new}, x_2^{new} \ldots x_n^{new}]$, where $x \in \mathbb{R}^{d_x \times 1}$, first, we used a moving horizon to divide it into several sequences as $X^{(new1)} = [x_1^{new}, x_2^{new}, x_3^{new}]$, $X^{(new2)} = [x_2^{new}, x_3^{new}, x_4^{new}], \ldots, X^{(newm)} = [x_{n-2}^{new}, x_{n-1}^{new}, x_n^{new}]$. Then, we put them into the GRU model and obtained the dynamic features $\{o^{(newi)}\}_{i=1}^{m}$ extracted by the GRU. Finally, the faults of the test samples are decided by the trained softmax regression model using the dynamic features.

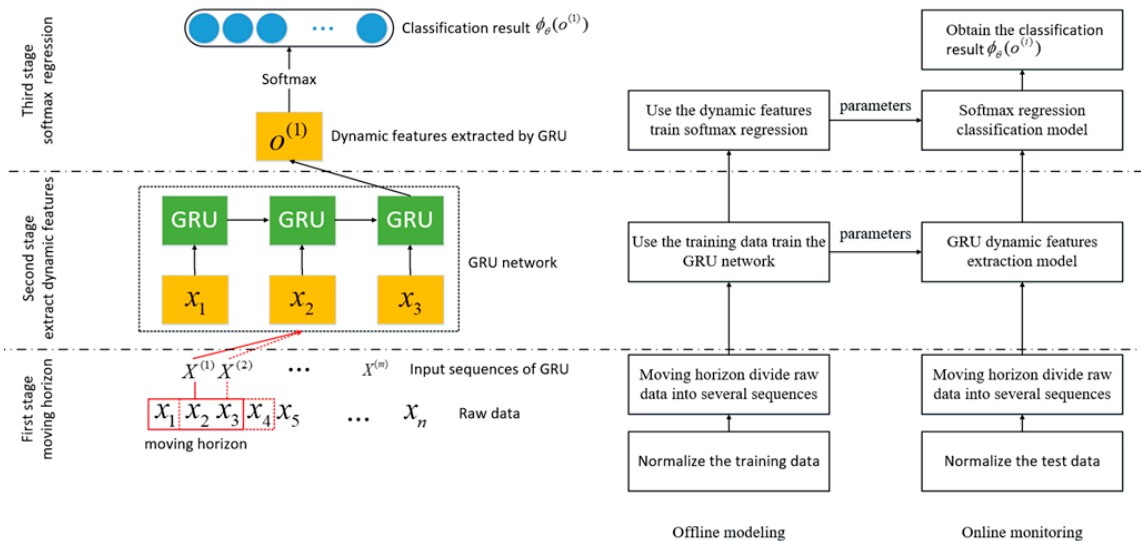

**Figure 4.** Flowchart of the three-stage fault diagnosis method based on GRU.

## 4. Case Study I: Fault Diagnosis of TE Using the Proposed Method

In this section, a GRU-based fault diagnosis algorithm is applied to the TE process, which is a benchmark case designed for testing the fault diagnosis performance. A model of this process was developed by Downs and Vogel [26], consisting of five major transformation units, which are a reactor, a condenser, a compressor, a separator, and a stripper, as shown in Figure 5. The MATLAB codes can be downloaded from http://depts.washington.edu/control/LARRY/TE/download.html. From this model, 41 measurements are generated along with 12 manipulated variables. A total of 21 different process upsets are simulated for testing the detection ability of the monitoring methods, as presented in Table 1 [27,28]. Our goal is to diagnose and classify the faults that have occurred, so normal data is not used as a training sample.

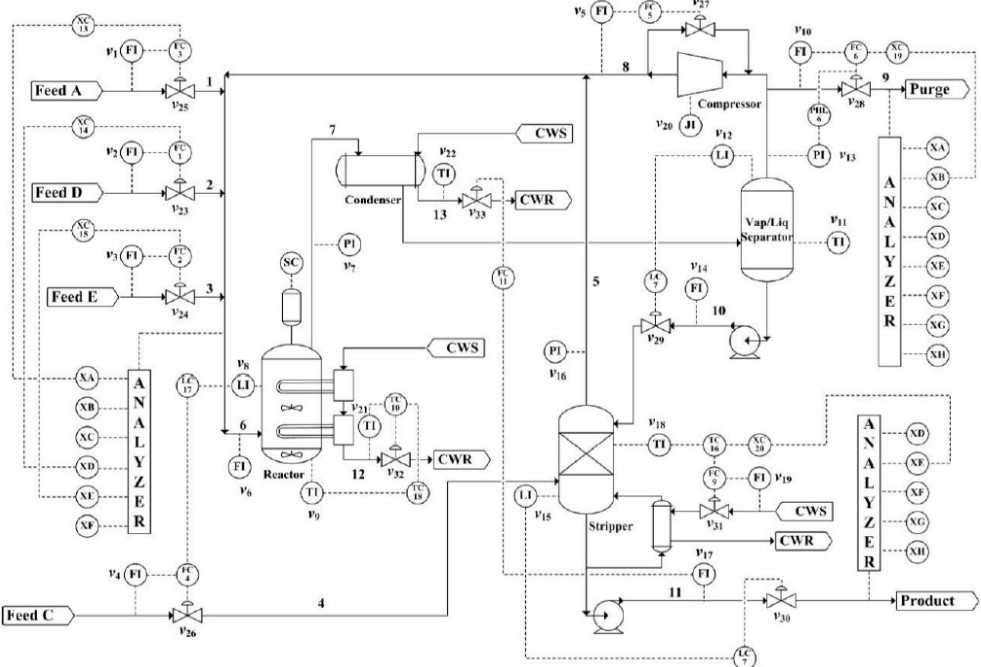

**Figure 5.** Flow diagram of the TE process. Reproduced with permission from Rato, T.J. and Reis, M.S., Chemometrics and Intelligent Laboratory Systems; published by Elsevier, 2013 [13].

The fault diagnosis algorithm in this paper is designed for time series problems or dynamic problems. We check whether the TE data has autocorrelation by calculating the autocorrelation coefficient of each variable of the TE data. The autocorrelation coefficient measures the degree to which the same event is correlated between two different periods. Suppose that the process has mean $\mu$ and variance $\sigma^2$ at time $t$. Then the definition of the autocorrelation between times $X_t$ and $X_{t+\tau}$ is:

$$R(\tau) = \frac{E[(X_t - \mu)(X_{t+\tau} - \mu)]}{\sigma^2} \tag{15}$$

where "$E$" is the expected value operator, $t$ is the lag, and $X_t (t = 1, 2 \ldots T)$. We selected a feature corresponding to the fault occurrence in the fault data of the TE process to calculate its autocorrelation between the time $X_t$ and $X_{t+\tau}$, $\tau = (1, 2 \ldots 20)$. The calculation results are shown in Figure 6. In the figure, approximate 95% confidence intervals are drawn with blue lines. From the results, this feature does have autocorrelation. Therefore, due to the recurrent structure and adaptive training strategy of the GRU, our proposed algorithm can fully extract the dynamic information in TE for further fault diagnosis.

**Table 1.** Process faults for the TE process simulator. Reproduced with permission from Rato, T.J. and Reis, M.S., Chemometrics and Intelligent Laboratory Systems; published by Elsevier, 2013 [13].

| Variable | Description | Type |
|---|---|---|
| IDV (1) | A/C feed ratio, B composition constant (Stream 4) | Step |
| IDV (2) | B composition, A/C ratio constant (Stream 4) | Step |
| IDV (3) | D feed temperature (Stream 2) | Step |
| IDV (4) | Reactor cooling water inlet temperature | Step |
| IDV (5) | Condenser cooling water inlet temperature | Step |
| IDV (6) | A feed loss (Stream 1) | Step |
| IDV (7) | C header pressure loss-reduced availability (Stream 4) | Step |
| IDV (8) | A, B, C feed composition (Stream 4) | Random variation |
| IDV (9) | D feed temperature (Stream 2) | Random variation |
| IDV (10) | C feed temperature (Stream 4) | Random variation |
| IDV (11) | Reactor cooling water inlet temperature | Random variation |
| IDV (12) | Condenser cooling water inlet temperature | Random variation |
| IDV (13) | Reaction kinetics | Slow drift |
| IDV (14) | Reactor cooling water valve | Sticking |
| IDV (15) | Condenser cooling water valve | Sticking |
| IDV (16)–IDV (20) | Unknown | Unknown |
| IDV (21) | The valve for Stream 4 was fixed at the steady state position | Constant position |

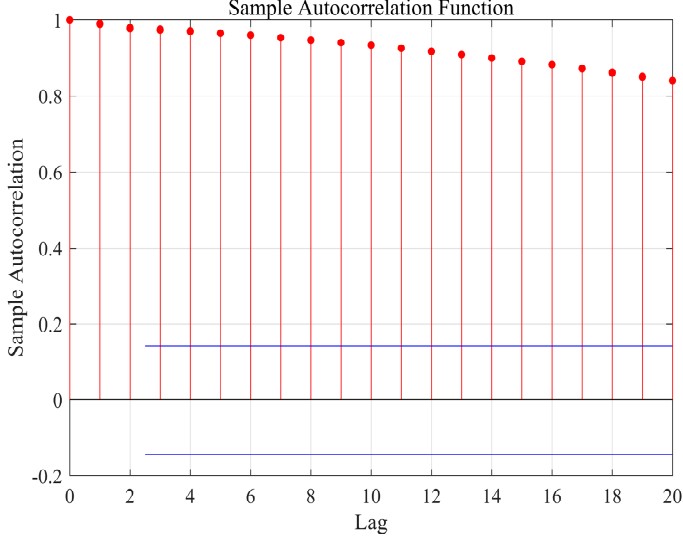

**Figure 6.** Autocorrelation charts of fault 1.

*4.1. Data Description*

The experimental dataset was generated by the TE simulation model, and 21 types of faults could be simulated. The simulation times of the training and the test sets were 24 h and 48 h, and the faults appeared after 1 h and 8 h, respectively. There were 480 sets of data for each fault as the training set. There were 800 sets for each fault as the test set. Since faults 3, 9, and 15 were difficult to diagnose with a data-based method, these three faults were not considered in our experiment. Therefore, there were a total of ($480 \times 18 = 8640$) sets training data and ($800 \times 18 = 14400$) sets of test data.

*4.2. Hyperparameters Selection and Fault Diagnosis Result and Analysis*

4.2.1. Hyperparameters Selection

Our GRU model contains two important hyperparameters: the number of GRU layers and the width of the moving horizon. We evaluated the accuracy for the GRU with different layers and different width of the moving horizon. The epochs of training were set to 30. Each accuracy was the result of averaging ten experiments, and the results are given in Table 2.

**Table 2.** The mean accuracy of the GRU with different layer numbers and different width of the moving horizon.

| Number of Layers | Width of the Moving Horizon | | | | | | | | |
|---|---|---|---|---|---|---|---|---|---|
| | 2 | 3 | 4 | 5 | 6 | 7 | 8 | 9 | 10 |
| 1 layer | 0.8236 | 0.8342 | 0.8342 | 0.8340 | 0.8302 | 0.8300 | 0.8295 | 0.8272 | 0.8254 |
| 2 layers | 0.8138 | 0.8228 | 0.8225 | 0.8214 | 0.8190 | 0.8163 | 0.8156 | 0.8106 | 0.8020 |
| 3 layers | 0.7984 | 0.8134 | 0.8123 | 0.8078 | 0.8070 | 0.8008 | 0.8006 | 0.7988 | 0.7985 |
| 4 layers | 0.7752 | 0.7880 | 0.7863 | 0.7849 | 0.7835 | 0.7803 | 0.7800 | 0.7762 | 0.7755 |

It is concluded from the table that when the number of GRU layers is set to one, and the width of the moving horizon is set to three and four, the accuracy reached a peak, but it decreased with the further increase of the width and the number of layers. The reason for this phenomenon was that as the number of GRU layers and the width of moving horizon increased, the amount of parameters, such as weights and biases in the model, was multiplied, which made the model's generalization ability worse and easy to overfit when dealing with high-dimensional industrial data.

Therefore, the network structure and hyperparameters are as follows: the number of GRU layers was set to 1, the width of the moving horizon was set to 3, the dimension of the hidden state $d_h$ was set to 30, and the parameters of batch normalization algorithm $\beta$ and $\varepsilon$ were set to 0. In the training, the mini-batch was set to 128, and the number of epochs was set to 30.

Experiments were run on a computer with Intel Core i7-7700 CPU, 8GB memory, and an NVIDIA GeForce GTX 1060 GPU. The diagnosis results of 21 faults are shown in a confusion matrix of Figure 7, where the confusion matrix considers target and output data. The target data are ground truth labels corresponding to 21 types of faults. The output data are the outputs from the tested method that performs classification. In the confusion matrix, the rows show the predicted class, and the columns show the ground truth. The diagonal cells show where the true class and predicted class match and the proportion. The off-diagonal cells show instances where the tested algorithm made mistakes and the proportion. The darker the color of the diagonal cell, the better the classification effect. Figure 6 has shown that only the diagnosis effect of fault 21 was not ideal, but the rest of the diagnosis results were satisfactory, with many fault diagnosis accuracy rates over 90%, and the mean accuracy was 87.36%.

In practical applications, we can collect online data during online monitoring and re-model and update parameters at regular intervals because the proposed model requires little computational cost and time cost. In this way, the diagnosis accuracy will be further improved. This is one of the advantages of the proposed model.

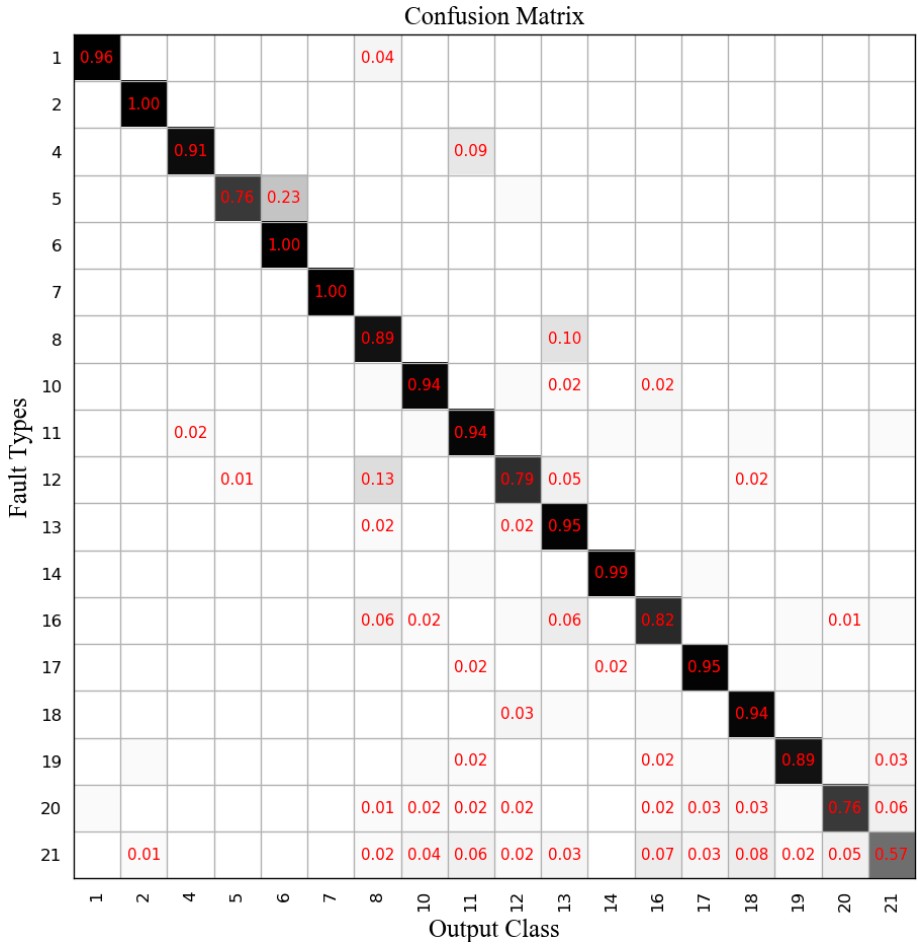

**Figure 7.** Confusion matrix of BN based GRU in 21 faults.

### 4.2.2. Effects of Batch Normalization

In deep learning, such as with a GRU, as the network deepens, there will be problems with the covariate shift, which will reduce the learning efficiency of the GRU network. The recently proposed batch normalization algorithm can effectively solve this problem. We can see the effect of batch normalization from the convergence speed and extent of the loss function during the training process in Figure 8. In addition, Table 3 compares the GRU and the BN-based GRU in several details and shows that the BN-based GRU is superior, both in terms of speed and accuracy.

The choice of model hyperparameters and the use of BN algorithms are theoretically based. Industrial data has the characteristic of being high-dimensional, and the deep network structure has too many parameters (weights and biases), thus it has poor generalization ability and is easily over-fitted when dealing with high-dimensional industrial data, that is, the "curse of dimensionality". Therefore, the GRU network is adopted in this paper. The GRU network is relatively sparse, so it has advantages in processing industrial data. The experimental results also show that the classification performance was superior when the number of layers and the width of moving horizon were both small. Moreover, in order to prevent over-fitting, the BN algorithm was cited herein to improve the GRU, and it turns out that the introduction of BN was effective. Consequently, the proposed method solves the "curse of dimensionality" in the industrial data to a certain extent.

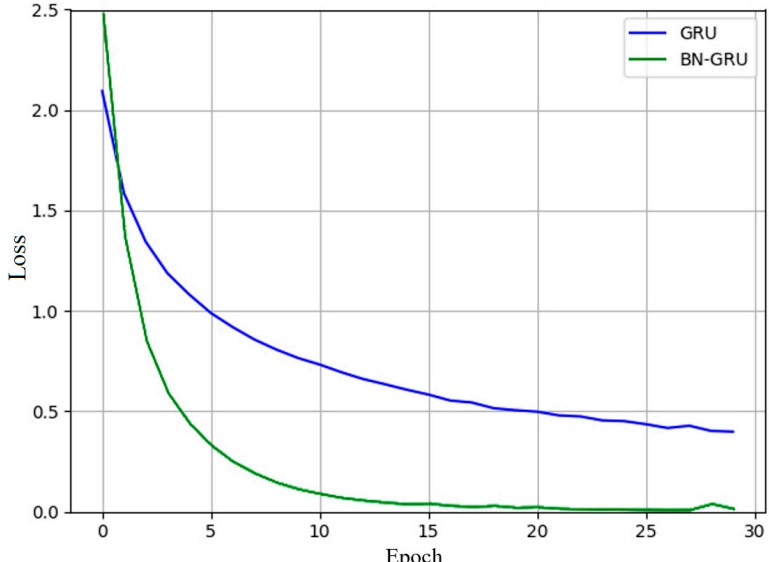

**Figure 8.** Loss function of GRU and BN based GRU during the training process.

**Table 3.** The comparison between GRU and BN-GRU.

| Details | GRU | BN-GRU |
|---|---|---|
| Training accuracy | 0.9429 | 0.9647 |
| Testing accuracy | 0.8342 | 0.8736 |
| Training loss | 0.3973 | 0.0012 |
| Testing loss | 0.7839 | 0.4493 |
| Epochs at convergence | 30 | 23 |

*4.3. Comparing with Related Work*

At the same time, we also conducted a comparative test. We used two fault diagnosis methods, DPCA-SVM and MLP, they both processed the sequence data to diagnose 21 faults also. According to the literature [7,13] and for the sake of fairness, the window size for DPCA was equal to the width of the moving horizon, which was equal to 3. For DPCA, we provided the performance under different reduced dimensions (the number of principal components) from 2 to 30. We also offered the performance of MLP and BN-based GRU under a different number of nodes in the hidden layer. The MLP used in this article is a five-layer network with the same number of nodes (dimensions) per layer. The diagnosis accuracy of the three methods in different cases is shown in Figure 9. The diagnosis accuracy shows that the proposed three-stage method based on a BN-based GRU can provide the best performance of all the methods.

We set the number of principal components of DPCA and the dimensions of the hidden layer in MLP and BN-based GRU equal to 30. The diagnosis results of DPCA-SVM are shown in Figure 10, and the mean accuracy rate was 66.40%. The diagnosis results of MLP are shown in Figure 11, where the mean accuracy rate was 77.23%.

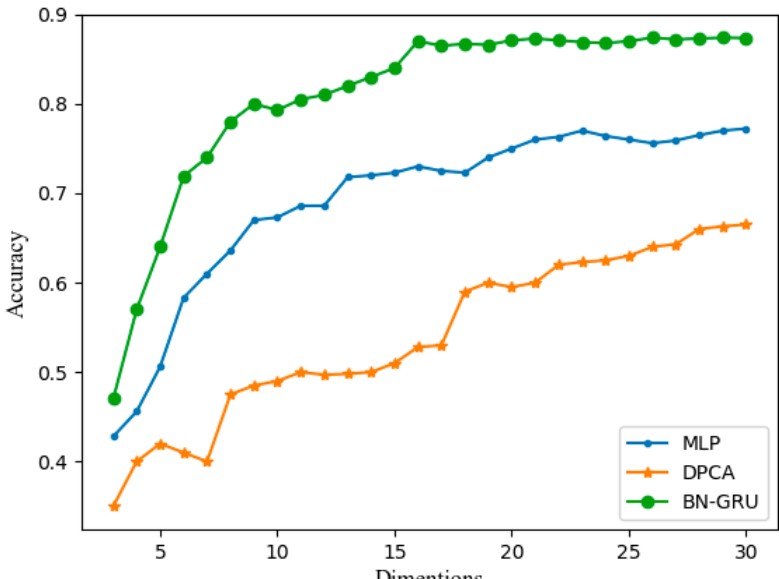

**Figure 9.** Diagnosis accuracy of DPCA-SVM, MLP, and BN-based GRU in different cases.

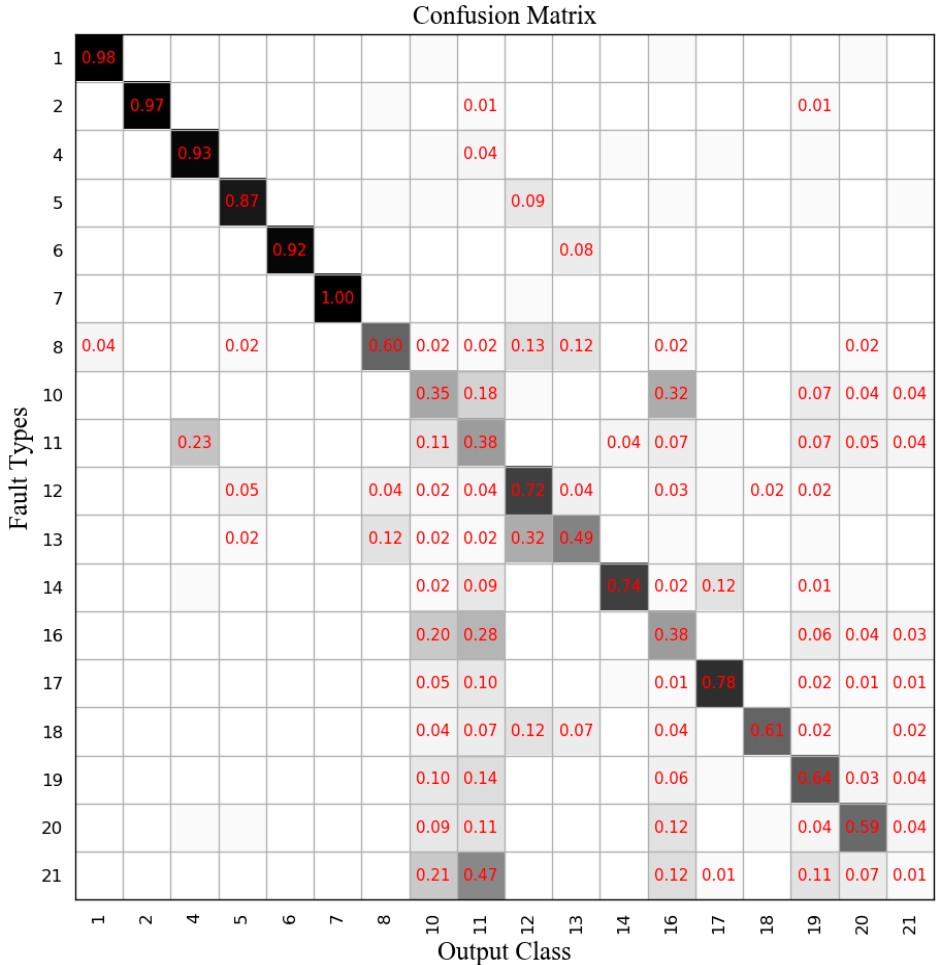

**Figure 10.** Confusion matrix of DPCA-SVM in 21 faults.

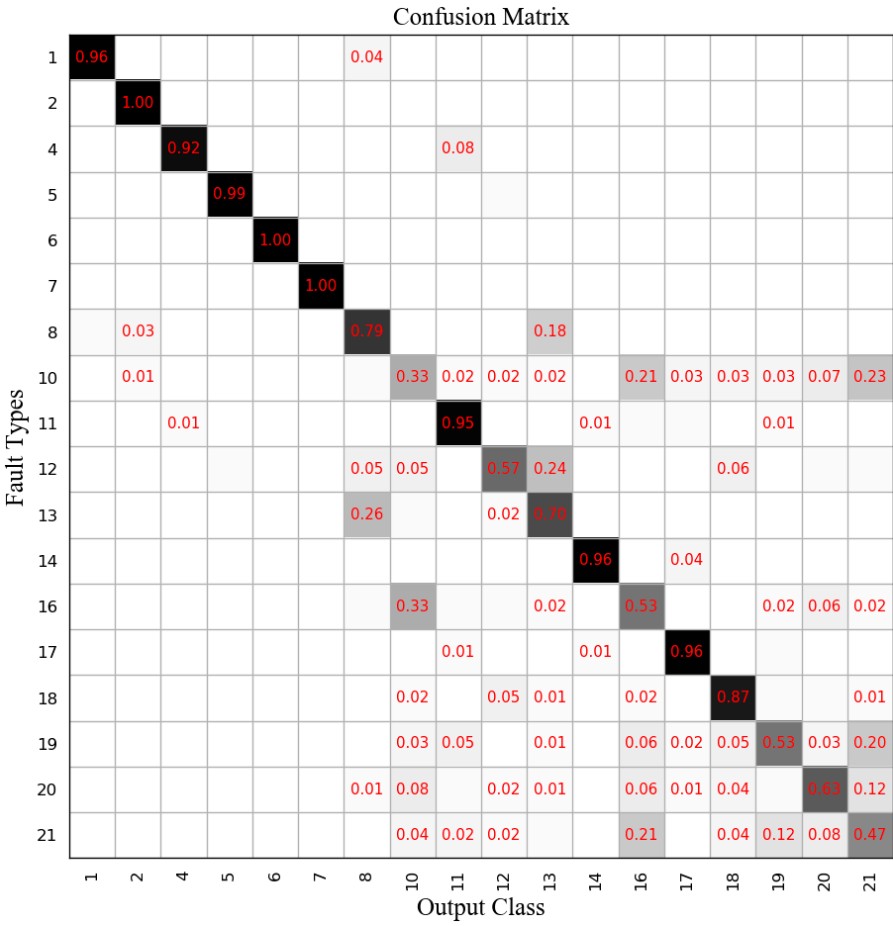

**Figure 11.** Confusion matrix of MLP in 21 faults.

We used the dimensionality reduction technique T-distributed stochastic neighbor embedding (t-SNE) to convert the features extracted by the three algorithms into a two-dimensional (2D) image, and the resulting scatter plot is shown in Figures 12–14. As shown in Figure 12, the feature extraction effect of DPCA was very poor, and only a few fault features were separated. The feature extraction effect of MLP was relatively good, and most of the fault features could be separated, but there were a few cases where, for example, faults 10, 19, 20, and 21 were confused. The fault extraction effect of a BN-based GRU was the best. Only a small part of the fault 20 and 21 were overlapping, and the rest of the features were well separated.

When dealing with data with small sizes (such as diagnosing certain types of TE faults), DPCA-SVM has considerable effects, but when dealing with large-scale data (such as diagnosing all 21 faults of TE), traditional methods like DPCA-SVM are not very effective. The GRU model of deep learning has a unique advantage in dealing with sequential data in the dynamic process. From the simulation results of the TE process, we could conclude that the proposed three-stage diagnosis method-based GRU in this paper was indeed superior to the traditional method.

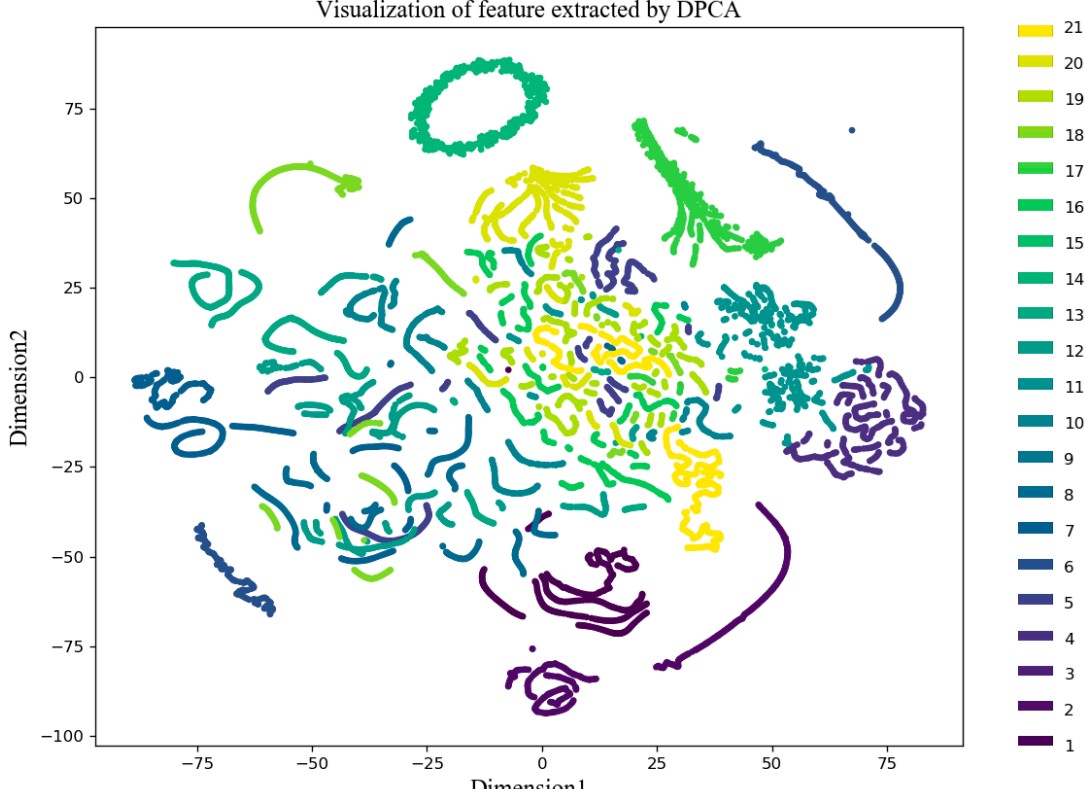

**Figure 12.** Visualization of features extracted using DPCA.

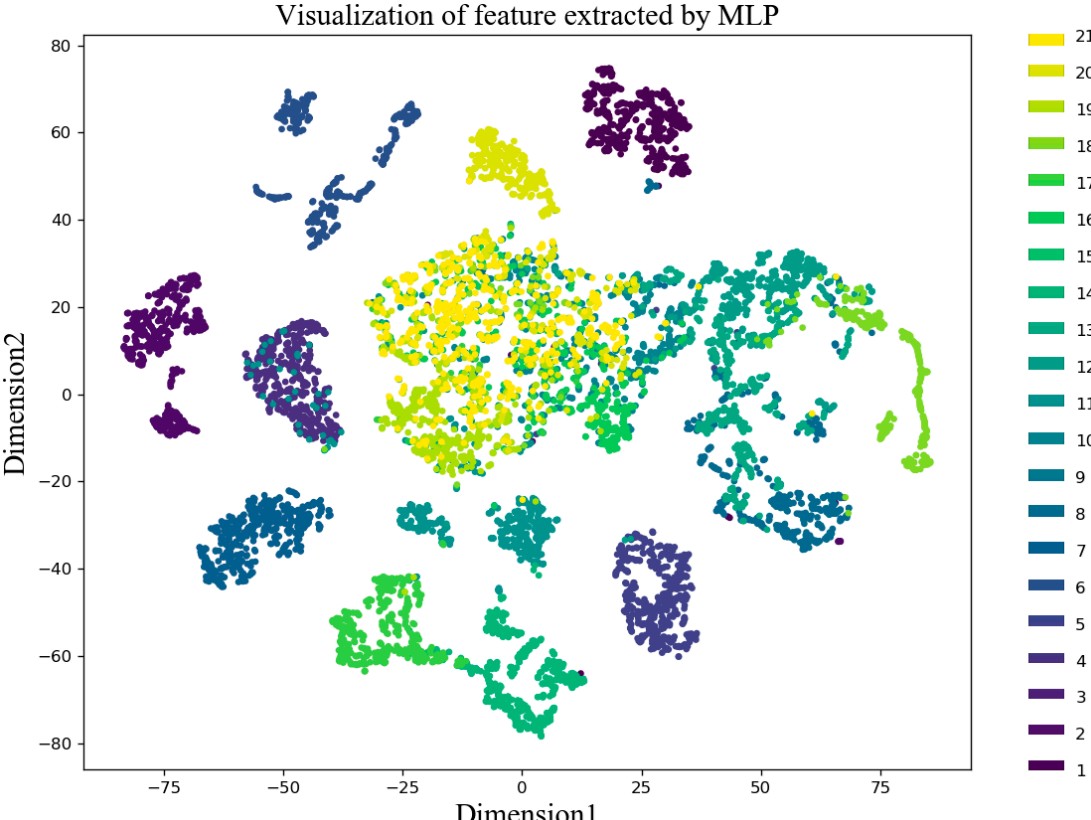

**Figure 13.** Visualization of features extracted using MLP.

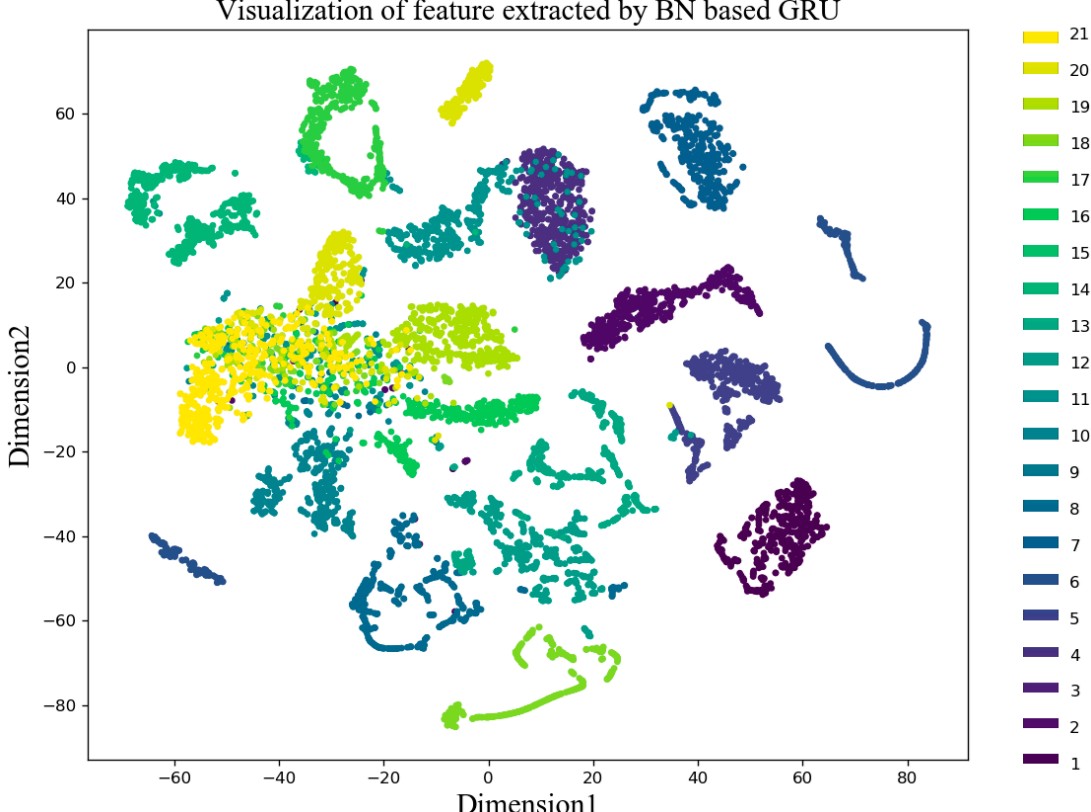

**Figure 14.** Visualization of features extracted using a BN-based GRU.

## 5. Case Study II: Fault Diagnosis of a PX Oxidation Process Using the Proposed Method

The PX oxidation reaction process is used for the production of PTA. There are three types of devices including one reactor, four condensers, and one reflux drum [29,30]. PX, acetic acid (solvent), cobalt acetate, manganese acetate (catalyst), tetrabromoethane (accelerator), and air were placed in the oxidation reactor to produce terephthalic acid (TA) in a high-temperature and high-pressure environment. [29,30]. The simplified flow chart of the PX oxidation process is shown in Figure 15. A total of nine different process upsets were simulated for testing the diagnosis ability of the proposed methods, as presented in Table 4.

**Table 4.** Process faults for the PX oxidation reaction process.

| Variable | Description | Type |
|----------|-------------|------|
| IDV (1) | Change of PX feed | Step |
| IDV (2) | Change of HAC feed | Step |
| IDV (3) | Change of H2O feed | Step |
| IDV (4) | Change of air feed | Step |
| IDV (5) | Change of PX feed temperature | Step |
| IDV (6) | A Step change of air feed temperature | Step |
| IDV (7) | Change of FC1102 temperature | Step |
| IDV (8) | Sticking of B1 valve | Sticking |
| IDV (9) | Sticking of condenser valve | Sticking |

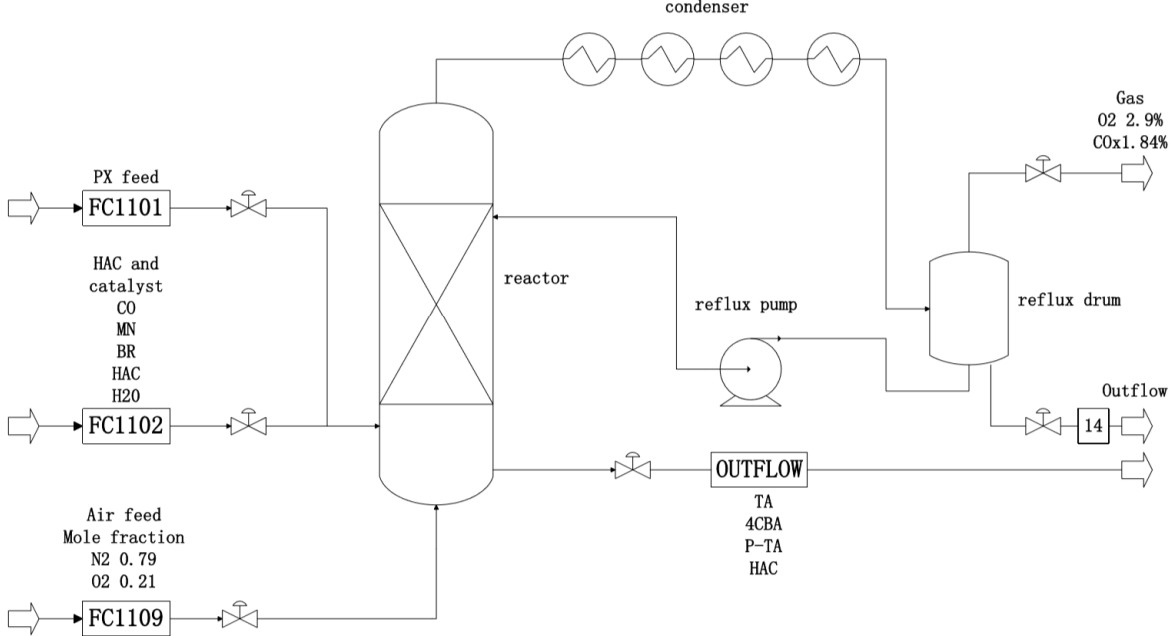

**Figure 15.** Simplified flow-chart of the PX oxidation process.

### 5.1. Data Description

The experimental dataset was collected by the PX oxidation process involving nine different fault types. The simulation times were 10 h, and the sampling frequency was 100 times per hour. There were 1000 sets of data for each fault as the dataset. Ten percent was used as a training set and the rest as a test set. Therefore, there were a total of ($100 \times 9 = 900$) sets of training data and ($900 \times 9 = 8100$) sets test data. The width of the moving horizon was also set to 10 in this experiment.

### 5.2. Fault Diagnosis Results and Analysis

In this experiment, the network structure and hyperparameters were as follows: the dimension of hidden state $d_h$ was set to 20, and the parameters of the batch normalization algorithm $\beta$ and $\varepsilon$ were set to 0. In training, the mini-batch was set to 32, the number of epochs was set to 30, the number of GRU layers was set to 1, and the width of the moving horizon was set to 3. The diagnosis results of nine faults are shown in a confusion matrix of Figure 16a. The visualization of feature extracted using a BN-based GRU is shown in Figure 16b. We can see that the dynamic information of the PX oxidation process data could be effectively utilized by the proposed method, and the mean testing accuracy reached 99.10%.

In actual industrial processes, labeled data is difficult to obtain. Therefore, in this experiment, we trained the network with very little data and got good results. This means that the proposed method can be applied to the fault diagnosis of dynamic processes in real industry.

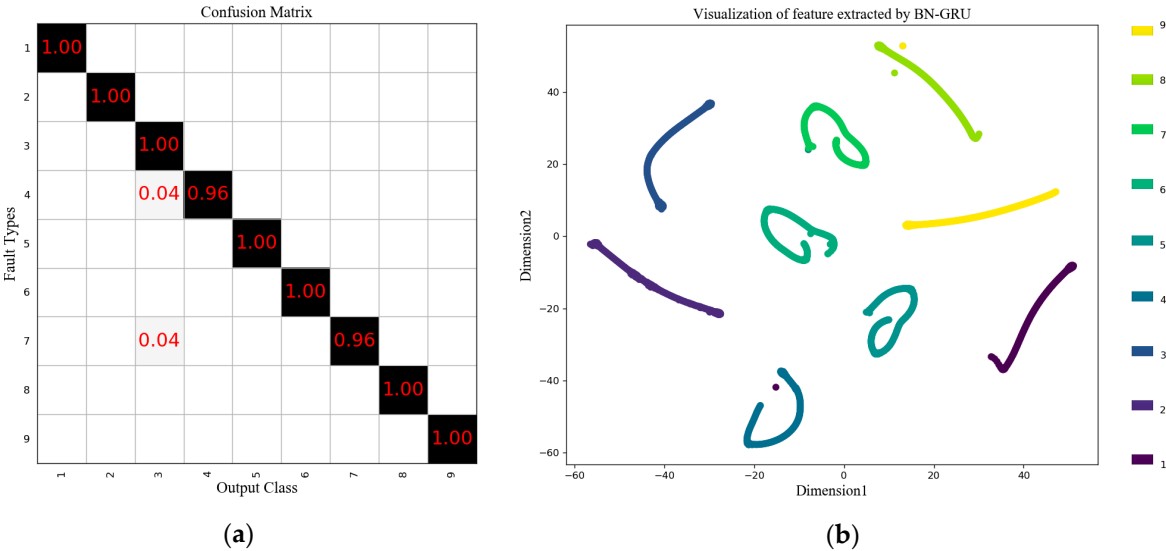

(a)

(b)

**Figure 16.** (**a**) The diagnosis results of PX oxidation process. (**b**) Visualization of features extracted using a BN-based GRU in the PX oxidation process.

## 5.3. Comparing with Related Work

In this case, the results of the proposed method are compared with two deep learning methods: DBN and CNN. In accordance with References [18,19], the neural numbers of DBN were set to $23 \times 20 \times 16 \times 9$, and the CNN consisted of a pair of convolutional layer and pooling layer with a convolution kernel size of 2. The diagnosis results of DBN are shown in Figure 17a, and the mean accuracy rate was 92.09%. The diagnosis results of CNN are shown in Figure 17b, where the mean accuracy rate was 97.56%. From the results, it can be clearly seen that the proposed method outperformed DBN and CNN in terms of the mean accuracy, showing the potential of the proposed GRU-based fault diagnosis method.

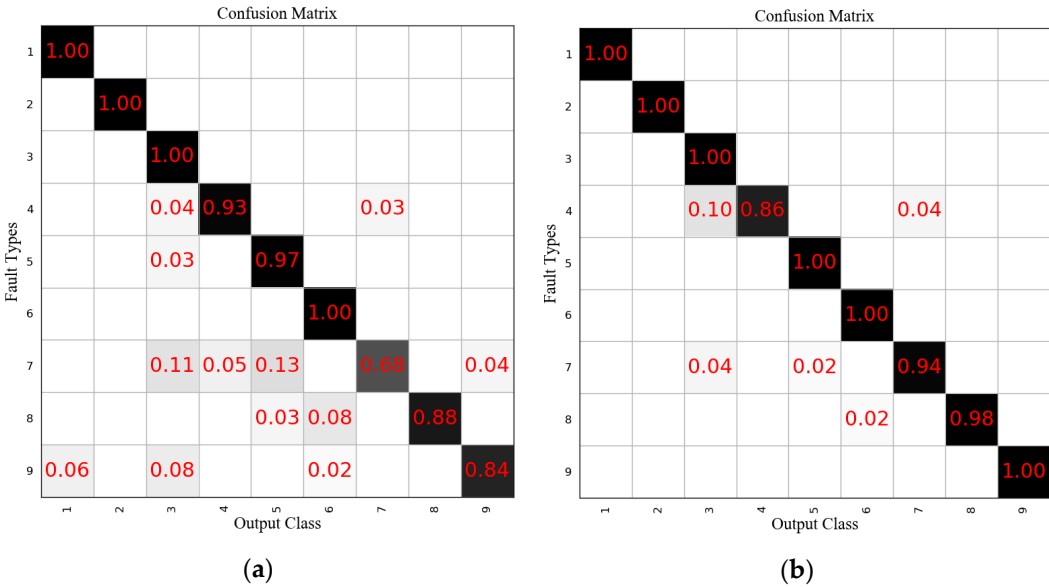

(a)

(b)

**Figure 17.** (**a**) The diagnosis results of the PX oxidation process on DBN. (**b**) The diagnosis results of the PX oxidation process on CNN.

*5.4. Practical Verification*

Due to the complexity of real industrial processes, the data collected is often not idealized. The existence of outliers in the training data should also be considered. In order to further verify the anti-interference ability and practicability of the proposed method, the outliers to the training data were added randomly in this case. We added 10 lots of fault 2 data in fault 1 and 10 lots of fault 5 data in fault 2 as well as fault 3.

As shown in Figure 18, only a small number of fault 3's were incorrectly classified as fault 5, and the diagnosis results of faults 1, 2, and 5 were unaffected. The mean accuracy remained at a high level with 98.93%. Consequently, the proposed method has practicality for real industrial processes.

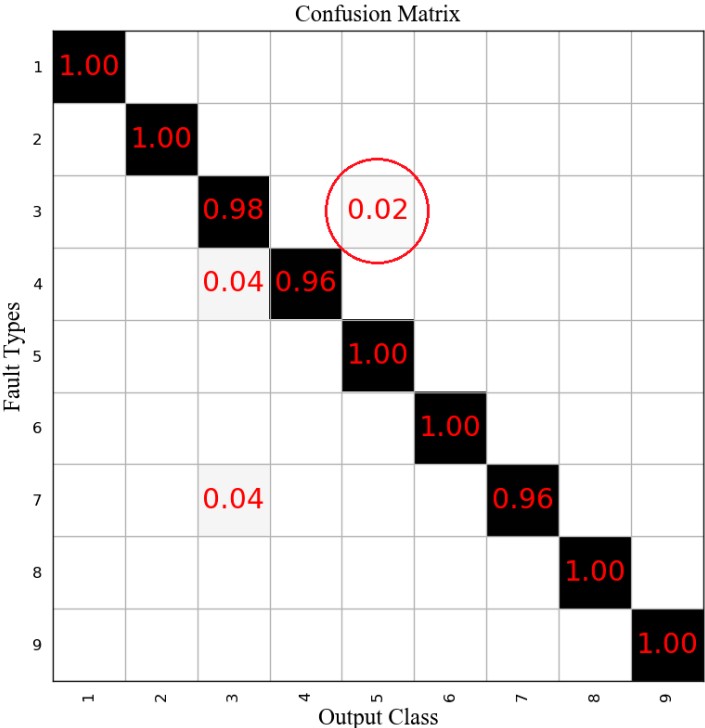

**Figure 18.** The fault diagnosis results with outliers in training data.

## 6. Conclusions

In this paper, a three-stage fault diagnosis method based on a GRU neural network was proposed. In this method, we used the moving horizon to process the sequence data in the industrial process and adjusted the time step by changing the width of the moving horizon. In this way, data could be better trained using the GRU neural network. Then, we trained the GRU neural network and optimized it with the BN algorithm to reduce the influence of the covariate displacement that existed in the deep learning. The GRU neural network was relatively simple and efficient, and it could guarantee both efficiency and high accuracy when extracting the dynamic features from sequential data. Finally, softmax regression gave an accurate probability interpretation of extracted dynamic features. By optimizing the hyperparameters of the network, the proposed method solved the "curse of dimensionality" in the industrial data to a certain extent. The simulation experiment of TE data and PX oxidation process data proved that the method could effectively extract the information in the dynamic process and improved the accuracy of fault diagnosis. In addition, online data during online monitoring can be collected to update model parameters, which will further improve the accuracy. In the future, this method can be applied to more complex industrial processes. Also, a further study on dynamic information in industrial process data will be put forward.

**Author Contributions:** The manuscript was conceptualized by both authors, where J.Y. developed the models, analyzed results, and wrote the manuscript; and Y.T. put forward the idea of this work, and contributed in model methodology and validation, and manuscript review and editing.

**Funding:** This research was funded by the Shanghai Sailing Program, grant number: 17YF1428300, and the Shanghai University Youth Teacher Training Program, grant number: ZZslg16009.

**Conflicts of Interest:** The authors declare no conflict of interest.

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
