# Peer review of "An Intelligent Fault Diagnosis Method Using GRU Neural Network towards Sequential Data in Dynamic Processes"

_processes, doi:10.3390/pr7030152_

Round 1

Reviewer 1 Report

An intelligent fault detection and isolation method is proposed using the GRU neural network, which is especially suitable for dynamic processes. I have the following comments:

- In terms of RNN, the depth of deep leaning means the number of layers, not the number of RNN units in the sequential direction. The influence of layer numbers can be studied as well.

- The comparison study can be more comprehensive. For example, DBN and CNN based methods have been proposed in the literature. Some may have even better results than the results in this article.

- Please provide the full name of “ANFIS” at its first appearance (line 42).

- How to choose the width of moving horizon? How about its influence?

- The symbol of Hadamard product is not shown correctly in this file.

Author Response

Response to Reviewer 1 Comments

 (Graphs and tables cannot be uploaded in the box, and the details are in the word file.)

Dear Reviewer,

We really appreciate all of your comments and suggestions. These comments are all valuable and very helpful for revising and improving our paper, as well as the important guiding significance to our researches. We have studied comments carefully and have made correction which we hope meet with approval. Revised portion are marked by the "Track Changes" function in Microsoft Word, so that changes are easily visible to the reviewer. The main corrections in the paper and the responds to the reviewer’s comments are as flowing:

Point 1: In terms of RNN, the depth of deep leaning means the number of layers, not the number of RNN units in the sequential direction. The influence of layer numbers can be studied as well.

Response 1: Thank you to pick out this point. It is really true as Reviewer suggested that both the number of GRU layers and the width of the moving horizon have an impact on the model. Combined with the fourth point, the number of GRU layers and the width of the moving horizon have been studied in lines 279 to 292. We used the "Track Changes" function in Microsoft Word, so that changes are easily visible to the reviewer, and the details are as follows:

Our GRU model contains two important hyperparameters the number of GRU layers and the width of the moving horizon. We evaluate the accuracy for the GRU with different layers and different width of the moving horizon. The epochs of training are set to 30. Each accuracy is the result of averaging ten experiments, and the results are given in Table 2.

It is concluded from the table that when the number of GRU layers is set to one, and the width of moving horizon is set to three and four, the accuracy reaches a peak, but it decreases with the further increase of the width and the number of layers. The reason for this phenomenon is that as the number of GRU layers and the width of moving horizon increase, the amount of parameters such as weights and biases in the model is multiplied, which makes the model's generalization ability worse and easy to overfit when dealing with high-dimensional industrial data.

Point 2: The comparison study can be more comprehensive. For example, DBN and CNN based methods have been proposed in the literature. Some may have even better results than the results in this article.

Response 2: Considering the Reviewer’s suggestion, the comparisons have been added in case study II from line 398 to 408 as follows:

5.3. Comparing with related work

In this case, the results of the proposed method are compared with two deep learning methods DBN and CNN. According to the literature [18] and [19], the neural numbers of DBN are set to  , and the CNN consists of a pair of convolutional layer and pooling layer with a convolution kernel size of 2. The diagnosis results of DBN are shown in Figure 16a, and the mean accuracy rate is 92.09%. The diagnosis results of CNN are shown in Figure 16b the mean accuracy rate is 97.56%. From the results, it can be clearly seen that the proposed method outperforms DBN and CNN in terms of the mean accuracy, showing the potential of the proposed GRU-based fault diagnosis method.

Point 3: Please provide the full name of “ANFIS” at its first appearance (line 42).

Response 3: The full name of “ANFIS” is adaptive neuro-fuzzy inference system’, and we have re-written this part in the article.

Point 4: How to choose the width of moving horizon? How about its influence?

Response 4: As shown in Response 1, we have added research on the width of moving horizon. when the number of GRU layers is set to one, and the width of moving horizon is set to three and four, the accuracy reaches a peak, but it decreases with the further increase of the width and the number of layers. The reason for this phenomenon is that as the number of GRU layers and the width of moving horizon increase, the amount of parameters such as weights and biases in the model is multiplied, which makes the model's generalization ability worse and easy to overfit when dealing with high-dimensional industrial data.

Point 5: The symbol of Hadamard product is not shown correctly in this file.

Response 5: We have corrected the symbol of Hadamard product and re-written the related formula according to the Reviewer’s suggestion.

Special thanks to you for your good comments.

Reviewer 2 Report

The authors developed gate recurrent unit (GRU) based process monitoring with three stages for correlated sequencial data in chemical process, where two case studies of TE and PX are compared.

My comments are as follows:

1) Because GRU is a kind of deep network framework, the curse of dimensionality problem will be the same in GRU method. Please explain how to solve COD issue in this GRU in the view point of # of weights and # ofhidden layers as well as # of data of training and test. This is an important issue of the contribution of this paper.

2) The structure of GRU network is important in any deep network. Please explain how to decide the structure of GRU network, layers, weights and algorithm as well as width of moving windows.

3) The monitoring result needs to be tested using real industrial data to show the performances of GRP in real data.

4) I'm wondering how to calculate the degree of correlation of extracted GRU output. Please show the correlation result of these GRU output to see how the GRU values are clearly reextracted from sequential dataset.

5) There are some mistyping from equation math and typo errors, which should be revised.

Author Response

Response to Reviewer 2 Comments

 (Graphs and tables cannot be uploaded in the box, and the details are in the word file.)

Dear Reviewer,

We really appreciate all of your comments and suggestions. These comments are all valuable and very helpful for revising and improving our paper, as well as the important guiding significance to our researches. We have studied comments carefully and have made correction which we hope meet with approval. Revised portion are marked by the "Track Changes" function in Microsoft Word, so that changes are easily visible to the reviewer. The main corrections in the paper and the responds to the reviewer’s comments are as flowing:

Point 1: Because GRU is a kind of deep network framework, the curse of dimensionality problem will be the same in GRU method. Please explain how to solve COD issue in this GRU in the view point of # of weights and # of hidden layers as well as # of data of training and test. This is an important issue of the contribution of this paper.

Response 1: Thank you to pick out this point. It is a great help for our work. It is really true that industrial data has the characteristics of high dimension, and the deep network structure has too many parameters (weights and biases), thus it has poor generalization ability and easy over-fitting when dealing with high-dimensional industrial data, that is, “curse of dimensionality”.

Experiment and analysis about hyperparameter are added in the manuscript. We used the "Track Changes" function in Microsoft Word, so that changes are easily visible to the reviewer. The experiment results show that the classification performance is superior when the number of layers and the width of moving horizon are both small. Moreover, in order to prevent over-fitting, the BN algorithm is cited herein to improve the GRU, and it turns out that the introduction of BN is effective. Consequently, the proposed method solves the “curse of dimensionality” in the industrial data to a certain extent. Considering the Reviewer’s suggestion, we have also made corresponding changes in the introduction and conclusion sections.

Point 2: The structure of GRU network is important in any deep network. Please explain how to decide the structure of GRU network, layers, weights and algorithm as well as width of moving windows.

Response 2: Thank you to pick out this point. It is really true as Reviewer suggested that both the number of GRU layers and the width of the moving horizon have an impact on the model. We have studied the number of GRU layers and the width of the moving horizon in lines 279 to 292, and the details are as follows:

Our GRU model contains two important hyperparameters the number of GRU layers and the width of the moving horizon. We evaluate the accuracy for the GRU with different layers and different width of the moving horizon. The epochs of training are set to 30. Each accuracy is the result of averaging ten experiments, and the results are given in Table 2.

It is concluded from the table that when the number of GRU layers is set to one, and the width of moving horizon is set to three and four, the accuracy reaches a peak, but it decreases with the further increase of the width and the number of layers. The reason for this phenomenon is that as the number of GRU layers and the width of moving horizon increase, the amount of parameters such as weights and biases in the model is multiplied, which makes the model's generalization ability worse and easy to overfit when dealing with high-dimensional industrial data.

Point 3: The monitoring result needs to be tested using real industrial data to show the performances of GRP in real data.

Response 3: As Reviewer suggested that the monitoring result needs to be tested using real industrial data. The parameters of the simulation model are adjusted based on real industrial systems to make our data as real as possible. We have tried our best to explore real industrial data, but due to conditions and financial constraints, we are temporarily unable to get it. If we are lucky enough to get real data in future work, we will give priority to it in the experiment.

Point 4: I'm wondering how to calculate the degree of correlation of extracted GRU output. Please show the correlation result of these GRU output to see how the GRU values are clearly reextracted from sequential dataset.

Response 4: This question stems from our controversial statement. The GRU calculates the dynamic feature of the data in the moving window to get the GRU output and classifies it, rather than calculating the degree of correlation of extracted GRU output. The proposed method is an end-to-end model. To prevent confusion with traditional methods, we have clarified this technical point and re-written the lines 53 to 65 of the manuscript.

Point 5: There are some mistyping from equation math and typo errors, which should be revised.

Response 5: We are very sorry for our incorrect writing. Mistyping from equation math and typo errors are revised in the manuscript.

Special thanks to you for your good comments.

Reviewer 3 Report

Authors describe the application of a Gate Recurrent Unit (GRU) network, together with a softmax classifier, to detect faults on temporal series coming from a set of distributed sensors. Proposed method is applied to the so-called Tennessee Eastman benchmark, as well as for an oxidation process. The proposed technique tries to substitute some already existing techniques for dealing with feature extraction in temporal series, where the dynamic nature of the process must be considered.

Authors claim that in this context, their proposal based on Gated Recurrent Units (GRU), as an improved variant of RNN, is able to maintain the temporal information, with a better performance in terms of learning. Dynamic features are applied, offline, by applying a moving horizon during the training process of the GRU.

The paper is well written and clearly organized, and it tackles the interesting problem of fault detection in multidimensional time series. However, the technical approach proposed by authors is not clear for this reviewer. Authors must clarify the technical points listed below, so it can be considered for publication:

-        Dynamic features are extracted offline, so adaptation is really achieved only for the training data set. How can authors guarantee that the dynamic of the temporal series is maintained during the whole lifetime of the system? This is important for a real system, not just for the model you are using.

-        Doing dynamic training offline (not online), can barely considered as being dynamic. You would need to recompute your dynamic features also online, but this would not allow retraining the classifier at the same time, since this is a supervised task. Do you have an alternative?

-        What happens if there exist outliers (faults) during training? In this case, your feature extraction will also consider outliers as part of the temporal trend of the curve.

-        The autocorrelation chart in Figure 6 shows that all samples are significant for the others, so all of them are correlated. This means that when a new measurement comes, dependencies are strong for time series estimation, so modelling is not possible in this temporal window. Do you preprocess your temporal series to remove this autocorrelation? (for instance, using differences instead of using the values). This is weird from the point of view of classical series forecasting, and so you should include a comment on this.

-        Comparisons in table 9 are provided for the same scenario (Tenesee), but based on implementations of the authors. However, this is a well-known problem in the literature, and many solutions have been also provided to tackle it. A real comparison with the results reported in the literature would be provided, instead of using baseline solutions created by authors. If you generate on your own your baseline solutions to be compared with, the credibility of the solution is reduced. Use values from the state-of-the-art instead. This prevents this reviewer to see the real quality of the proposed solution, and the validity it will have at the end for the community.

Author Response

Response to Reviewer 3 Comments

 (Graphs and tables cannot be uploaded in the box, and the details are in the word file.)

Dear Reviewer,

We really appreciate all of your comments and suggestions. These comments are all valuable and very helpful for revising and improving our paper, as well as the important guiding significance to our researches. We have studied comments carefully and have made correction which we hope meet with approval. Revised portion are marked by the "Track Changes" function in Microsoft Word, so that changes are easily visible to the reviewer. The main corrections in the paper and the responds to the reviewer’s comments are as flowing:

Point 1: Dynamic features are extracted offline, so adaptation is really achieved only for the training data set. How can authors guarantee that the dynamic of the temporal series is maintained during the whole lifetime of the system? This is important for a real system, not just for the model you are using.

Response 1: Thank you for picking out this point. As Reviewer comments that it is important to guarantee the dynamic of the temporal series. The proposed method is only for the dynamic process in this paper, and the models we used are typical representatives of these real systems. In practical applications, the training data must be broad and sufficient enough to ensure that the model is well trained. And the raw data is divided into several sequence units by the moving horizon; the condition is satisfied when the dynamics of the data in each sequence unit is guaranteed. This step enhances the practicality of the proposed method.

Point 2: Doing dynamic training offline (not online), can barely considered as being dynamic. You would need to recompute your dynamic features also online, but this would not allow retraining the classifier at the same time, since this is a supervised task. Do you have an alternative?

Response 2: Thank you to pick out this point. It is really true as Reviewer comments that feature extraction and classification both affect the diagnosis performance but are designed individually in traditional machine learning models (Eg pca+svm). This is a divide and conquer strategy which cannot be optimized simultaneously. The proposed method is an end-to-end deep learning model. That is to say, the feature extraction algorithm and the classifier are trained based on historical data and labels at the same time. Deep learning can learn the abstract representation features of the raw data automatically, which could avoid the requirement of prior knowledge. This is an advantage of deep learning over traditional methods.

We have clarified this technical point and re-written the lines 53 to 65 of the manuscript. We used the "Track Changes" function in Microsoft Word, so that changes are easily visible to the reviewer

Point 3: What happens if there exist outliers (faults) during training? In this case, your feature extraction will also consider outliers as part of the temporal trend of the curve.

Response 3: Training data is important for deep learning models, to ensure model accuracy, there should be as few outliers as possible in training data. Outlier detection methods can be applied to filter the training data to ensure the reliability of it.

Point 4: The autocorrelation chart in Figure 6 shows that all samples are significant for the others, so all of them are correlated. This means that when a new measurement comes, dependencies are strong for time series estimation, so modelling is not possible in this temporal window. Do you preprocess your temporal series to remove this autocorrelation? (for instance, using differences instead of using the values). This is weird from the point of view of classical series forecasting, and so you should include a comment on this.

Response 4: Thank you to pick out this point. Due to the chain structure, RNN models have the ability to take advantage of the correlation among data. That is to say, the proposed method can also process autocorrelation data without preprocessing the temporal series, which is the advantage of the proposed method over the conventional method. We have clarified the difference between classical methods and deep learning and re-written the lines 53 to 65 of the manuscript according to the Reviewer’s comments.

Point 5: Comparisons in table 9 are provided for the same scenario (Tenesee), but based on implementations of the authors. However, this is a well-known problem in the literature, and many solutions have been also provided to tackle it. A real comparison with the results reported in the literature would be provided, instead of using baseline solutions created by authors. If you generate on your own your baseline solutions to be compared with, the credibility of the solution is reduced. Use values from the state-of-the-art instead. This prevents this reviewer to see the real quality of the proposed solution, and the validity it will have at the end for the community.

Response 5: It is really true as Reviewer comments that comparison should be based on relevant literature. There are many types of faults in TE processes. Different types of faults are analyzed in different literature (for example, fault 1 2 4 6 8 12 15 are analysed in the literature [13]). This paper covers 2 to 30 when selecting the number of DPCA principal components and the number of MLP hidden layer nodes. The baseline in the literature [7] [13] are included. According to the reviewer's suggestion, the relevant literature [7] [13] is marked in the article. And considering the Reviewer’s suggestion, we have added the comparison and provided the results reported in the relevant literature in case study II from line 398 to 408 as follows:

5.3. Comparing with related work

In this case, the results of the proposed method are compared with two deep learning methods DBN and CNN. According to the literature [18] and [19], the neural numbers of DBN are set to , and the CNN consists of a pair of convolutional layer and pooling layer with a convolution kernel size of 2. The diagnosis results of DBN are shown in Figure 16a, and the mean accuracy rate is 92.09%. The diagnosis results of CNN are shown in Figure 16b the mean accuracy rate is 97.56%. From the results, it can be clearly seen that the proposed method outperforms DBN and CNN in terms of the mean accuracy, showing the potential of the proposed GRU-based fault diagnosis method.

Round 2

Reviewer 1 Report

I have no further comments.

Author Response

Dear Reviewer,

thanks again to you for your good comments and patience.

Reviewer 3 Report

In my opinion, the answers provided by authors to my concerns in the points discussed below are still not satisfactory:

Regarding points 1) and 3), authors claim that "the training data must be broad and sufficient enough to ensure that the model is well trained." This is clear, but after reading their comments it seems that their approach is valid because they work with a training dataset with good features, but which is far from being generalizable or applicable in real works, beyond this dataset. This would be the case for any work relying on the dataset, but it's critical in their case because authors claim they offer a dynamic solution, which in reality only considers dynamic nature of time series offline, while keeping the fault detection performed online, as a static process. As stated in Point 2, this assumes, at the end, that by evaluating your system during an initial partial period (learning stage), you are able to guarantee that the dynamic nature of the system is integrated in your Deep Learning Model, but once again, it cannot cope with changes during system lifetime, that is where the term dynamic would make sense. As an example, would you considered DPCA as dynamic if Principal components are only computed during training offline?

Regarding point 5), my comment was not to provide more experimental results on your own, but to take the results already reported in the papers in the state of the art and to compare them with your proposal. Moreover, could you provide more information on images in Figure 11?

Author Response

Dear Reviewer,

We appreciate all of your further comments and suggestions. It is a great help for our work and my personal research skills. We tried our best to improve the manuscript and made some changes in the manuscript. The main corrections in the paper and the responds to the reviewer’s comments are as flowing:

Point: Regarding points 1) and 3), authors claim that "the training data must be broad and sufficient enough to ensure that the model is well trained." This is clear, but after reading their comments it seems that their approach is valid because they work with a training dataset with good features, but which is far from being generalizable or applicable in real works, beyond this dataset. This would be the case for any work relying on the dataset, but it's critical in their case because authors claim they offer a dynamic solution, which in reality only considers dynamic nature of time series offline, while keeping the fault detection performed online, as a static process. As stated in Point 2, this assumes, at the end, that by evaluating your system during an initial partial period (learning stage), you are able to guarantee that the dynamic nature of the system is integrated in your Deep Learning Model, but once again, it cannot cope with changes during system lifetime, that is where the term dynamic would make sense. As an example, would you considered DPCA as dynamic if Principal components are only computed during training offline?

Regarding point 5), my comment was not to provide more experimental results on your own, but to take the results already reported in the papers in the state of the art and to compare them with your proposal. Moreover, could you provide more information on images in Figure 11?

Response: Considering the Reviewer’s suggestion, we have experimented with outliers in the training data to further verify the anti-interference ability and practicability of the proposed method. The experiment results have helped us a lot in our work. The modified content is in the line 419 to 429. The details are as follows:

5.4 Practical verification

Due to the complexity of real industrial processes, the data collected is often not idealized. The existence of outliers in the training data should also be considered. In order to further verify the anti-interference ability and practicability of the proposed method, the outliers to the training data are added randomly in this case. We add 10 fault 2 data in fault 1 and 10 fault 5 data in fault 2 as well as fault 3.

As shown in Figure17, only small-scale of faults 3 were incorrectly classified as fault 5, and the diagnosis results of fault 1, 2, 5 were unaffected. The mean accuracy remains at a high level with 98.93%. Consequently, the proposed method has practicality in the real industrial process.

The parameters of the simulation model are adjusted based on real industrial systems to make our data as real as possible. The training dataset is also selected according to the real system. This question stems from our controversial statement. I apologize for my unclear comments.   

"Dynamics" is embodied in two aspects of our approach. One is the dynamic processing of data by the moving horizon, which is similar to the step of constructing augmented matrices in DPCA. The second point is that we can collect online data during online monitoring and re-model and update parameters at regular intervals. Because our model requires little computational cost and time cost, this is one of the advantages of our model. The reviewer's comments remind us that we have omitted the detailed explanation of the second point. We have clarified this technical point in line 318 to 321as well as line 447 to line 448.

The other reviewer pointed out that we need to provide a comparison experiment. According to your opinion, we have provided the results reported in the papers in the state of the art and to compare them with my proposal. And related literature is also marked in Case Study I.

Due to t-SNE parameter error, Figure 11 is a little wrong but does not affect the conclusion and structure of the article. We took the initiative to correct it.

Special thanks to you for your good comments and patience.
